# Synthesis of a Rare Water-Soluble Silver(II)/Porphyrin and Its Multifunctional Therapeutic Effect on Methicillin-Resistant *Staphylococcus aureus*

**DOI:** 10.3390/molecules27186009

**Published:** 2022-09-15

**Authors:** Jiaqi He, Yu Yin, Yingjie Shao, Wenkai Zhang, Yanling Lin, Xiuping Qian, Qizhi Ren

**Affiliations:** 1School of Chemistry and Chemical Engineering, Shanghai Jiao Tong University, Shanghai 200240, China; 2School of Pharmacy, Shanghai Jiao Tong University, Shanghai 200240, China

**Keywords:** bivalent silver complex, sulfonated porphyrin, photodynamic therapy, photocatalytic NADH oxidation, multifunctional effect, methicillin-resistant *Staphylococcus aureus*

## Abstract

Porphyrin derivatives are popular photodynamic therapy (PDT) agents; however, their typical insolubility in water has made it challenging to separate cells of organisms in a liquid water environment. Herein, a novel water-soluble 5,10,15,20-tetrakis(4-methoxyphenyl-3-sulfonatophenyl) porphyrin (TMPPS) was synthesized with 95% yield by modifying the traditional sulfonation route. The reaction of TMPPS with AgNO_3_ afforded AgTMPPS an unusual Ag(II) oxidation state (97% yield). The free base and Ag(II) complex were characterized by matrix-assisted laser desorption ionization-mass spectroscopy, and ^1^H nuclear magnetic resonance, Fourier-transform infrared, UV-vis, fluorescence, and X-ray photolectron spectroscopies. Upon 460 nm laser irradiation, AgTMPPS generated a large amount of ^1^O_2_, whereas no ⦁OH was detected. Antibacterial experiments on methicillin-resistant *Staphylococcus aureus* (MRSA) revealed that the combined action of Ag^Ⅱ^ ions and PDT could endow AgTMPPS with a 100% bactericidal ratio for highly concentrated MRSA (10^8^ CFU/mL) at a very low dosage (4 μM) under laser irradiation at 360 J/cm^2^. Another PDT response was demonstrated by photocatalytically oxidizing 1,4-dihydronicotinamide adenine dinucleotide to NAD^+^ with AgTMPPS. The structural features of the TMPPS and AgTMPPS molecules were investigated by density functional theory quantum chemical calculations to demonstrate the efficient chemical and photodynamical effects of AgTMPPS for non-invasive antibacterial therapy.

## 1. Introduction

Clinical antibacterial treatment has become increasingly difficult because of the resistance of pathogenic bacteria to antibiotics [1,2]. Owing to the multi-drug resistance (MDR) property of methicillin-resistant *Staphylococcus aureus* (MRSA), it has become difficult to apply conventional antibiotic options available for the treatment of such infections. Meanwhile, vancomycin, the first-line drug of choice for invasive MRSA infections for many years, does not provide effective long-term protection against the resurgence or reinvasion of resistant organisms [3,4]. Typically, research on a novel antibiotic requires extensive human effort and a high cost investment over decades before clinical application. However, pathogens can develop drug resistance within two weeks [5], leading to difficulties with the development of antibiotics [6]. The likelihood of fatality in patients with MRSA is estimated to be 64% higher than that in patients with non-resistant types of infections, which implies a poor clinical outcome [4,7]. Therefore, it is necessary to develop therapeutic strategies with high antimicrobial efficiency and a low risk of drug resistance to effectively eliminate MRSA bacteria.

Silver can effectively inhibit the proliferation of pathogenic bacteria with a broad spectrum, and there is extensive literature on the antibacterial effects of Ag^+^ [8,9,10,11]. Although the specific antibacterial mechanism of silver-based materials has not been entirely elucidated, several conclusions have been arrived at by the scientific community with widespread acceptance. One among them is that silver ions (Ag^+^) can act on various targets, including bacterial cell membranes; bind to and inhibit thiol-containing proteins; and release reactive oxygen species through several processes [12]. However, Ag^+^ is a moderate oxidizing agent (ϕ_Ag+/Ag_^0^ = 0.80 V), and only the most sensitive redox sites are affected by it [13]. There are increasing reports of microbial resistance to silver compounds [14,15], necessitating the development of more powerful formulations of silver with a higher oxidizing potential.

Ag^Ⅱ^ ions have a higher redox potential (1.987 V) than Ag^+^, and have acquired attention to overcome the aforementioned limitations [13]. The higher reduction potential implies higher biological potential energy. Thus, a more efficient antibacterial effect can be achieved by Ag^2+^. This unusual higher oxidation state can be obtained by the destruction of the quasi-closed shell with the d^10^ electronic configuration; however, it leads to difficulties with stabilization [16]. Recently, N-donor ligands have been developed to maintain the high valence state of silver [17], offering a novel route for antibacterial therapy by the irreversible chemical oxidization of various cellular components.

Porphyrins are rigid tetrapyrrole macrocycles and can serve as ligands with double anions after losing their inner protons. Porphyrin ligands can form complexes with almost all metals from the periodic table, and present four N atoms in a square-planar arrangement to metal ions located at the center [18]. Porphyrin derivates, such as porfimer sodium as an anti-tumor drug, have long been employed as photodynamic therapy (PDT) agents owing to their considerable photodynamic effects, superior photophysical properties, and low side effects in humans [19,20]. Furthermore, transition metal coordination complexes have been found to possess excellent stability owing to the effect of chelates and macrocycles. Recently, transition metalloporphyrin complexes (e.g., Pd^Ⅱ^ [21,22], Zn^Ⅱ^ [23], and Co^Ⅱ^ [24]) have been studied as photosensitizers for the PDT of pathogenic bacteria as a novel method to control antibiotic-resistant infections [25]. Most laboratory-synthesized porphyrins are water-insoluble [26]. However, the cells of organisms cannot be separated from the liquid water environment, and the hydrophobic nature can induce the aggregation of photosensitizers in water via π-π interactions. To attain the desired therapeutic effect, it is necessary to design novel effective and water-soluble porphyrins for PDT [27].

We have been working on the photocatalytic and biological applications of water-soluble sulfonated porphyrins and metalloporphyrins [28,29], as well as an iridium(III)-porphyrin sonosensitizer for the sonodynamic therapy of tumors [30]. Herein, we report the synthesis of high-valent silver(II) 5,10,15,20-tetrakis(4-methoxyphenyl-3-sulfonatophenyl) porphyrin (AgTMPPS) containing water-soluble sulfonated ligands. The complex was highly soluble in water, and the central silver stably existed in the form of Ag^2+^. The systematic antibacterial experiments using AgTMPPS against MRSA were performed to evaluate sterilization activity; and we also revealed the antibacterial mechanism. Overall, this work presents an efficient PDT strategy with a novel silver(II) porphyrin and highlights it capabilities for non-invasive bacterial sterilization with metal ion and photodynamic effects.

## 2. Results and Discussion

### 2.1. Preparation and Characterization of the Silver-Based Porphyrin

The synthesis of AgTMPPS is depicted in Figure 1a. As per the literature method [30], TMPP was first fabricated by mixing pyrrole and *p*-methoxy benzaldehyde in propionic acid, following which TMPPS was prepared by sulfonation with concentrated H_2_SO_4_. Finally, to anchor the silver ions onto the functionalized organic ligands, AgNO_3_ was added to initiate a classic coordination reaction, thus forming the metalloporphyrin AgTMPPS. The complex AgTMPPS was further purified using a dialysis bag (MWCO = 1 K). Figure 1b shows the solubility in the H_2_O/CH_2_Cl_2_ mixing system. TMPPS and its silver complex dissolve well in water owing to the presence of sulfonate groups [31]. The measured solubility of AgTMPPS was 0.1771 mol/L. Homogeneous metallization in water is a convenient synthesis strategy and offers a high yield [32,33]. Figure 1c shows the centrifuged precipitate. The precipitate appears to be mirror-like silver, which is confirmed to be elemental silver by X-ray photoelectron spectroscopy (XPS, Appendix A). Another equivalent of AgNO_3_ is disproportionated into Ag(II) of a higher valence. The tetrapyrrole cavity of metalloporphyrin allows transition metal ions to exist in an unusual oxidation state [34].

The sulfonated porphyrin TMPPS and its silver-based complex were analyzed by matrix-assisted laser desorption ionization-time of flight mass spectroscopy (MALDI-TOF MS), ^1^H nuclear magnetic resonance (NMR), UV-vis absorption spectroscopy, fluorescence emission spectroscopy, Fourier-transform infrared (FT-IR) spectroscopy, and X-ray photoelectron (XPS) spectroscopy (Figure 2). 

Owing to the multiple cleavage reactions of water-soluble porphyrins, excimer ions were used to confirm the target product during MS. The highest peaks in the MALDI-TOF mass spectra for TMPPS and AgTMPPS are located at 1055.27 and 1161.35 (*m*/*z*), respectively, indicating the successful coordination of silver ions on the porphyrin cavity in the AgTMPPS sample (Figure 2a). The ^1^H NMR spectra are shown in Figure 2b. There are six types of protons in TMPPS, including NH-pyrrole, -OCH_3_, 5-phenyl, 6-phenyl, 2-phenyl, and β-pyrrole at −2.86, 4.09, 7.46, 8.20–8.28, 8.41–8.65, and 8.86 ppm, respectively. Accurate ^1^H NMR spectra for AgTMPPS could not be acquired due to the paramagnetic nature of the silver(Ⅱ) porphyrin caused by the d^9^ ion configuration of silver ions (Appendix A). Nevertheless, the peak at −2.86 ppm, which is the characteristic peak from pyrrolic-NH, completely disappears after silver coordination, confirming the successful synthesis of AgTMPPS. 

The incorporation of metal ions into the porphyrin ring can change the properties of the porphyrin [33]. Figure 2c shows the UV-vis absorption spectra for TMPPS and AgTMPPS in water. For the free-base porphyrin, the Soret peak at 417 nm and four Q bands at 520, 562, 582, and 650 nm were observed, which arise from a_1u_(π)→e_g_(π*) and a_2u_(π)→e_g_*(π) transitions, respectively [35]. When a delocalized π bond is formed between pyrrolic N atoms and silver ions, the increased average electron cloud density on the porphyrin ring could lower the electron transition energy [36]. Hence, the Soret absorption of AgTMPPS exhibits a strong broadband absorption and red shift to 429 nm. Simultaneously, the peak numbers of the Q bands range from 4 to 2 due to the change in the entire molecular symmetry from D_2h_ to D_4h_ after the insertion of silver [37]. In their fluorescence emission spectra (Figure 2d), two typical emission peaks were observed at 654 and 710 nm for TMPPS, with a fluorescence lifetime of 7.85 ns (Appendix A). However, AgTMPPS undergoes complete fluorescence quenching because the introduction of a metal into the porphyrin ring can affect the fluorescence emission properties, and the mixing of the metallic d_π_ orbital with the π* orbital on the porphyrin ring leads to a loss of the fluorescence signal [35]. Furthermore, the metal ion with the partially filled d orbital can quench fluorescence by electron or energy transfer, and the paramagnetic d^9^ valence electron configuration has been shown to increase the quenching efficiency.

The FT-IR spectra are shown in Figure 2e and Appendix A. Several strong characteristic absorptions assigned to υ(-SO_2_) can be observed at 1091, 1161, and 1188 cm^−1^ for AgTMPPS and at 1093, 1151, and 1191 cm^−1^ for TMPPS. Moreover, the N-H bending vibration from TMPPS at 974 cm^−1^ disappears after coordination with silver [38,39]. These results support the synthesis of AgTMPPS.

The XPS profiles (Figure 2f and Figure 3) provide more detailed information on the two porphyrins, and especially confirm the existence of Ag(II) ions in AgTMPPS. The XPS survey scan of AgTMPPS shows clear Ag, O, N, C, and S signals (Figure 2f). The oxidation state of coordinated metal ions is heavily influenced by the given ligand environment. In the Ag 3d spectrum, two characteristic peaks were observed at 368.4 and 374.4 eV (Figure 3a), corresponding to those of the standard Ag(Ⅱ) oxide (368.2 and 374.2 eV), and confirming that the silver ions in AgTMPPS exist in a divalent state. The slightly increased binding energy of 0.2 eV implies a decrease in the electron cloud density of Ag(II) ions due to the conjugated porphyrin macrocycle. The different chemical properties of the two porphyrins are reflected in the N 1s spectrum (Figure 3b). The curve for TMPPS can be fitted to two characteristic peaks at 397.8 and 399.9 eV, corresponding to -NH- and -C=N- [40]. In contrast, the curve-fitted N 1s spectrum of AgTMPPS shows only one peak at 399.5 eV, which is the characteristic peak for the four equivalent pyrrolic N atoms of the metalloporphyrin [41]. Thus, all the above results confirm the successful synthesis of Ag(Ⅱ)/porphyrin complexes.

### 2.2. Photodynamic Performance of the Silver-Based Porphyrin

As previously reported, porphyrin-based materials exhibit strong optical absorption and efficient photodynamic capacity [42,43]. To confirm the excitative photodynamic effect of the porphyrin-derived complex (AgTMPPS) under 460 nm irradiation, a 9,10-anthracenediyl-bis(methylene) dimalonic acid (ABDA) probe was employed to intuitively measure the singlet oxygen in solution [44], which could be oxidized to the corresponding endoperoxide (ABDAO_2_) by ^1^O_2_. The changes in the UV-vis spectra for ABDA and AgTMPPS over time were detected under laser irradiation and without irradiation (460 nm, 0.05 W cm^−2^). Figure 4a shows the changes in the curves for the 378 nm absorption peak of ABDA with increasing irradiation dose. The singlet oxygen sensor probe exhibited a decrease in absorbance. The absorbance of the ABDA photo-oxidized by AgTMPPS reached a minimum upon 30 min of laser irradiation, the rate constant for the oxidation of ABDA is 2.2 × 10^−2^ min^−1^ (Appendix A), and the conversion rate is 56.65%. In contrast, no change was observed in the absorption curves of the non-irradiated group, for which the rate constant and conversion rate are 2.9 × 10^−4^ and 0.90%, respectively (Appendix A). Furthermore, the methylene blue probe was selected as the reference standard, and the calculated singlet oxygen yield of AgTMPPS was 67.33%, as shown in Appendix A. Thus, the Ag-doped porphyrin-derived complex can generate numerous singlet oxygens under 460 nm visible-light irradiation, providing many possibilities for inducing oxidative stress to kill bacteria using PDT. To further investigate other reactive oxygen species (ROS, e.g., •OH), methylene blue (MB) was used for monitoring during the same laser irradiation process (460 nm, 0.05 W/cm^2^; Figure 4b). MB can be oxidatively degraded by •OH in solution [45], but there is no distinct absorption peak change with increasing irradiation time. Hence, AgTMPPS cannot generate hydroxyl radicals (•OH) under 460 nm light irradiation. One disadvantage of the porphyrin derivative is that it easily photodegrades under continuous exposure to light, hindering its efficient and durable application [46]. The photostability of the Ag-based porphyrin in water was further investigated using UV-vis spectra (Figure 4c and Appendix A). The intensities of the Soret bands of AgTMPPS remain the same under light irradiation, suggesting that the material is stable during PDT, thereby highlighting its potential to achieve antibacterial effects.

Electron paramagnetic resonance (EPR) spectra were acquired to further validate ^1^O_2_ and •OH. 2,2,6,6-Tetramethylpiperidine (TEMP) and 5,5-dimethyl-1-pyrroline-N-oxide (DMPO) were used to scavenge the singlet oxygen and hydroxyl radicals, respectively [47]. The blue line in Figure 4d indicates an increased three-line signal between 3470 and 3520 G when the AgTMPPS/TEMP solution was subjected to 460 nm irradiation, signifying the capture of ^1^O_2_. However, the spectrum of the non-irradiated AgTMPPS + TEMP group exhibited no radical signal (yellow line). Moreover, the signal of the DMPO adduct produced by •OH was not observed without and after light irradiation (red and green lines, respectively). Thus, AgTMPPS can produce ^1^O_2_ without •OH after laser irradiation.

### 2.3. Density Functional Theory Computation of the Silver-Based Porphyrin

Porphyrin-based molecules can generate ^1^O_2_ from the surrounding ^3^O_2_ when they are excited to the triplet state by light [48,49]. Hence, we studied the electronic structures using the density functional theory (DFT). For all atoms except silver, which used the LANL2DZ basis set, geometry optimization was carried out at the B3LYP/6-311G(d) level. The electron cloud densities of the highest occupied molecular orbital (HOMO) and lowest occupied molecular orbital (LUMO) for TMPPS were primarily concentrated over the cyclic tetrapyrrole (Figure 5a). However, the charge distribution on the frontier molecular orbitals changed after silver complexation. The HOMO of AgTMPPS was primarily localized on the macrocycle and the silver ion, and the LUMO shifted toward the periphery of the system (Figure 5b). This more-separated orbital distribution suggests the possibility of a smaller energy gap [50]. The calculated HOMO and LUMO energy gaps of the two porphyrins are 2.75 (TMPPS) and 2.55 eV (AgTMPPS), respectively, indicating that AgTMPPS is easier to transform to the excited state after light irradiation, which is associated with its strong capability to generate ROS [51]. The calculated geometric configuration of AgTMPPS (Figure 5) demonstrates that the Ag^Ⅱ^ ion is coordinated in a near-ideal square-planar fashion. The smaller oxidized d^9^ ion fits the porphyrin cavity well and experiences crystal-field stabilization effects [18].

DFT primarily helps researchers understand an experiment by providing important physical information. In this study, DFT quantum chemical calculations were conducted (Table 1) to investigate the fluorescence quenching of AgTMPPS. We focused on the electronic transition to calculate the possible fluorescence lifetime (τ) [52]. A radiative lifetime less than 10 ns corresponds to the emissive state, and the electronic transition of the free porphyrin from S_0_→S_4_ results in fluorescence. In contrast, for a fluorescence lifetime longer than 10 ns, the corresponding state becomes dark due to the long relaxation time for the excited electron, which is caused by the nonradiative process during the electronic transitions of AgTMPPS. These calculated results are consistent with the results from the fluorescence emission spectra (Figure 2d).

### 2.4. Photocatalytic Oxidation of NADH by the Silver-Based Porphyrin

The endogenous 1,4-dihydronicotinamide adenine dinucleotide (NADH), as a coenzyme in numerous oxidoreductase reactions, helps maintain the intracellular redox balance [53]. Consequently, the selective initiation of NADH oxidation can damage the respiratory metabolism response by disturbing the redox homeostasis [54]. We investigated whether AgTMPPS could act as a photocatalyst to catalyze the oxidation of NADH and provide a novel pathway of the photo-oxidation effect. UV-vis absorption spectroscopy was performed to monitor the NADH oxidation process photocatalyzed by AgTMPPS under laser irradiation over time, and the NADH + Laser and NADH + AgTMPPS groups were used as a control. The presence of an extra hydrogen of the planar pyridine ring in NADH leads to significant spectral differences between NADH and NAD^+^ (Appendix A). The band at 339 nm originates from the n→π* transition of the dihydronicotinamide part, which is in an electron conjugation coplanar conformation with the carboxamide moiety [55]. The spectrum of NAD^+^ does not exhibit the 339 nm band due to a twisted and cis conformation between the oxidized nicotinamide and carboxamide [56]. The absorption intensity at 339 nm for NADH decreases greatly due to the action of AgTMPPS upon laser irradiation, and the rate constant is 6.6 × 10^−3^ min^−1^ (Figure 6a,b). However, no clear changes are observed in the spectra of the control groups, and the oxidation rate constants are 4.5 × 10^−4^ and 1.6 × 10^−4^ min^−1^ for NADH + AgTMPPS and NADH + Laser, respectively (Appendix A). The oxidation turnover number (TON; the mole ratio of NAD^+^/AgTMPPS) was calculated (Figure 6c) to measure the catalytic efficiency. The TONs are 0.11 and 1.35 when catalyzed by AgTMPPS without and with laser irradiation, respectively, suggesting the promising photocatalytic action of AgTMPPS by the oxidation of NADH. Furthermore, the absorbance at 260 nm (assigned to the adenine ring) is intense, suggesting that adenine is not destroyed.

To further monitor the laser-excited oxidation of NADH catalyzed by AgTMPPS, the ^1^H NMR spectra in D_2_O/CD_3_OD (1/3, *v*/*v*) are shown in Figure 6d. The oxidization of dihydronicotinamide to nicotinamide led to a considerable change in the chemical shifts of other protons on the nicotinamide rings. The ^1^H NMR spectrum shows new peaks at 8.32, 8.55, 8.99, 9.35, and 9.55 ppm upon 10 min of laser irradiation, assignable to the protons at the nicotinamide ring of NAD^+^ [57]. Thus, NADH was converted into NAD^+^. However, no significant change was observed without 460 nm laser irradiation or AgTMPPS.

### 2.5. In Vitro Antibacterial Performance of the Silver-Based Porphyrin against MRSA

Influenced by the antibacterial properties of the Ag^Ⅱ^-ion complex, the high efficiency of ^1^O_2_ generation, and the photocatalytic oxidation of NADH, systematic antibacterial experiments were performed to evaluate the sterilization performance of AgTMPPS. Gram-positive MRSA, the strain that is resistant to all ꞵ-lactam antibiotics [58], was selected as the model bacteria. The experimental schematic is shown in Appendix A. The minimum inhibitory concentration (MIC) and minimum bactericidal concentration (MBC) for solutions of the silver-based complex and silver nitrate are shown in Table 2 for different light doses and under a power density of 50 mW/cm^2^. For the non-irradiated groups (Table 2, entry 1), the MIC/MBC values of Ag^Ⅱ^TMPPS and Ag^Ⅰ^NO_3_ were 16/64 and 32/128 μmol/L, respectively, while for TMPPS, they were both ˃2048 μmol/L. Hence, TMPPS itself has a limited ability to inhibit bacterial growth without light irradiation, and the Ag^Ⅱ^-inserted porphyrin complex exhibits better antibacterial properties than Ag^Ⅰ^NO_3_ owing to the enhanced bacterial destructive effect of bivalent silver ions. Note that for AgTMPPS without irradiation, such a sterilization approach via Ag^Ⅱ^ ions cannot completely eradicate bacteria; further, the MBC (64 μmol/L) was much higher than the MIC (16 μmol/L), suggesting that the bacteria soon breed and return to their original quantity. Therefore, laser irradiation, which can promote ROS generation and NADH oxidization, was performed to overcome this problem (Table 2, entrys 2–4). With increasing light dose under 460 nm laser irradiation, the MIC for AgNO_3_ did not change, whereas the MBC decreased slightly and remained at 64 μmol/L, implying that the antibacterial effect of the silver ions is independent of light. However, the MIC and MBC of AgTMPPS decreased considerably, and a minimum value of 4/8 μmol/L was obtained, indicating that a combined chemical and photodynamic effect can achieve bacterial eradication. 

To further evaluate the antibacterial activity of AgTMPPS, we performed irradiation at a dose of 360 J/cm^2^ in the subsequent agar plate counting analysis. Similar antibacterial activities were obtained by the spread plate method (Figure 7a). Figure 7b shows the colony counts, and Figure 7c shows the corresponding calculated sterilization rates. We conducted evaluations for the control, group subjected to only saline treatment, and group with no antibacterial effect; the results show that the saline group subjected to light irradiation exhibited almost no change (Table 3, entrys 1 and 6). Without light irradiation and with increasing concentration from 0.5 to 4 μmol/L, AgTMPPS showed limited bacterial inhibition, and the sterilization rates gradually increased from 1.71 to 55.29% (red columns in Figure 7c, entries 2–5 in Table 3). Similar experimental evaluations were conducted under 460 nm light irradiation and a 360 J/cm^2^ irradiation dose. The sterilization rates changed to 50.14, 70.29, 97.82, and 100% at concentrations from 0.5 to 4 μmol/L (blue columns in Figure 7c, entries 7–10 in Table 3). Thus, AgTMPPS at 2 and 4 μmol/L exhibited excellent MRSA-killing ability (97.82 and 100%) due to the combined effects of the silver ions and photodynamic-based dual-antibacterial therapy.

The results show that AgTMPPS possesses excellent multifunctional antibacterial ability, highlighting its potential for bacterial disinfection in the biological field (Appendix A).

## 3. Materials and Methods

### 3.1. Materials and Instruments

Propionic acid (≥99.5%), pyrrole (99.0%), ethanol (C_2_H_5_OH, ≥99.8%), dichloromethane (CH_2_Cl_2_, ≥99.9%), ABDA (≥95.0%), MB (≥98.0%), and NADH (98.0%) were purchased from Aladdin Biochemical Technology Co., Ltd., Shanghai, China, Sulfuric acid (H_2_SO_4_, ≥96.0%), fuming sulfuric acid (20%), *p*-methoxy benzaldehyde (≥99.0%), silver nitrate (AgNO_3_, ≥99.9%), sodium carbonate (Na_2_CO_3_, ≥99.8%), sodium hydroxide (NaOH, ≥99.9%), Luria-Bertani (LB) agar, and LB broth were purchased from Sinopharm Chemical Reagent Co., Ltd., Shanghai, China, The experimental consumables were mostly purchased from Corning (Corning, NY, USA) or BioFil (JET, Guangzhou, China).

The MALDI-TOF mass spectra were measured using an Autoflex Speed TOF/TOF instrument (Bruker, Karlsruhe, Germany) under the positive charge mode. Proton NMR (^1^H NMR) spectra were acquired using an AVANCE Ⅲ HD 500 instrument (Bruker, Karlsruhe, Germany). FT-IR spectra were recorded on a Nicolet 6700 spectrometer (Thermo Fisher, MA, USA) in the range of 400–4000 cm^−1^ using KBr tablets. XPS profiles were obtained using an XPS system (AXIS Ultra DLD, Kratos, Manchester, UK) to investigate the chemical state of the samples. The UV-vis absorption spectra were recorded on a Shimadzu UV-1800 spectrophotometer (Kyoto, Japan). The fluorescence emission spectra were detected using an FLS 1000 system (Edinburgh Instruments, Edinburgh, UK). 

### 3.2. Preparation of TMPP

First, 37.4 mL of *p*-methoxy benzaldehyde (0.31 mol) and 500 mL of propionic acid were heated to 130 °C in a four-necked flask. Subsequently, 20.9 mL of pyrrole (0.30 mol) was dissolved in propionic acid (100 mL), and the solution was injected into the system within 30 min. The mixture was refluxed for another 40 min and cooled naturally to 80 °C. Thereafter, 200 mL of C_2_H_5_OH was added, and the mixture was allowed to stand at 4 °C overnight. The solid product was filtered and washed with ethanol until the filtrate was colorless. After drying, blue-purple crystals were obtained (yield: 28%).

### 3.3. Preparation of TMPPS 

First, 1.20 g of TMPP (1.63 mmol) in 150 mL of a CH_2_Cl_2_ solution was transferred to a flask, and was then gradually injected in 10 mL of a CH_2_Cl_2_ mixture containing concentrated H_2_SO_4_ (3.0 mL, 98%) and fuming H_2_SO_4_ (0.75 mL, 20%). The reaction system was stirred at 25 °C overnight and the solid was separated out. The solid was dissolved with deionized water and repeatedly washed with CH_2_Cl_2_ to remove the unsulfonated porphyrins. The pH of the aqueous phase was adjusted to 8–9 using a saturated Na_2_CO_3_ solution, and then filled in a dialysis bag (MWCO = 1 K). The dialysis solution was magnetically stirred in 3000 mL of deionized water, and the water was changed every hour (7 times) to remove free ions. After drying under vacuum, the purple product TMPPS was obtained in a yield of 95%. ^1^H NMR (500 MHz, DMSO-*d*_6_): *δ* 8.86 (s, 8H, *ꞵ*-pyrroles), 8.65–8.41 (m, 4H, 2-phenyl), 8.28–8.20 (m, 4H, 6-phenyl), 7.46 (dd, *J* = 8.7, 3.0 Hz, 4H, 5-phenyl), 4.09 (s, 12H, -OCH_3_), −2.86 (s, 2H, N-H pyrrole). MALDI-TOF MS (positive) m/z: 1055.27 ([M + H]). FT-IR spectra (KBr) cm^−1^: υ(-SO_2_) [1191, 1151, 1093], δ(N-H)-pyrrole [974], υ(C-H) [2943, 2842], υ(C=C)-phenyl [1601, 1491], υ(C-H) + δ(C-H)-phenyl [901, 799], π(C-H)-pyrrole [830], υ(C-O)-phenyl-O-C [1256]. UV-vis absorption spectra (ultrapure water) λ: 650, 582, 562, 520, 417 nm. Fluorescence emission spectra (ultrapure water; λ_ex_ = 420 nm) λ: 654, 710 nm.

### 3.4. Synthesis of Ag^Ⅱ^TMPPS (High-Valent Silver Complexes of TMPPS)

First, 104.2 mg of TMPPS (0.09 mmol) was dissolved in deionized water (20 mL) in a flask, and then adjusted to pH 8–9 with a NaOH solution (0.1 mol/L). Subsequently, 600.0 mg of AgNO_3_ (3.53 mmol) was added at 110 °C, and the solution was refluxed for 6 h under continuous stirring and refluxing; throughout the process, the reaction system was maintained in a weakly basic state. The reaction was continued for 6 h. After cooling to 25 °C, the mixture was centrifuged (8000 rpm, 5 min, 6 times) to discard the solid precipitate. Finally, uncoordinated Ag^+^ was removed by dialysis (MWCO = 1 K, 5 h, 6 deionized H_2_O exchanges), and the solution was spin-dried in a vacuum. The resulting red-brown Ag^Ⅱ^TMPPS product was obtained at a yield of 97%. MALDI-TOF MS (positive) m/z: 1161.35 ([M + H]). FT-IR spectra (KBr) cm^−1^: υ(-SO_2_) [1188, 1161, 1091], υ(C-H) [2921, 2848], υ(C=C)-phenyl [1596, 1494], υ(C-H) + δ(C-H)-phenyl [912, 806], π(C-H)-pyrrole [845], υ(C-O)-phenyl-O-C [1262]. UV-vis absorption spectra (ultrapure water) λ: 580, 544, 429 nm.

### 3.5. ROS Generation in Solution 

The generation of ^1^O_2_ and ⦁OH by AgTMPPS in solution under laser irradiation was measured by the ABDA probe and MB, respectively. For the experiment with ABDA, the typical probe (100 µmol/L in DMSO) was mixed with an aqueous solution of AgTMPPS (10 µmol/L) in a quartz cell. After laser irradiation (460 nm, 50 mw/cm^2^) for various durations, the UV-vis spectra were measured, and the decomposition rate was quantified via absorbance changes at 378 nm; 20 µmol/L MB was used to track the ⦁OH radicals, and mixed with AgTMPPS (10 µmol/L) in ultrapure water. Subsequently, a 460 nm laser beam was focused on the solution at a power density of 50 mw/cm^2^ for different irradiation durations. At the same time, the absorption was recorded by a UV-vis spectrophotometer.

### 3.6. EPR Assay

To detect the radicals of ^1^O_2_ and ⦁OH, EPR measurements were carried out (EMXplus-9.5/12, Bruker, Karlsruhe, Germany) at ambient temperature under 460 nm laser irradiation. The sample was contained in an NMR tube positioned in a resonant cavity with an illumination window. The EPR parameters were sweep widths of 800.0 G and 200.0 G, a time constant of 10.24 ms, and a conversion time of 20.02 ms, yielding a sweep time of 120.12 ms. Field modulation was applied at 100.00 kHz and 2.000 G, and the microwave attenuation was 13.0 dB (~10.02 mW). The spin traps, TEMP (for trapping ^1^O_2_, 80 mmol/L) and DMPO (for trapping ⦁OH, 90 mmol/L), were used to verify the formation of the generated radicals by the action of the complex (5 mmol/L).

### 3.7. Photocatalytic Oxidation of NADH

AgTMPPS was used for the catalytic oxidation of NADH under laser irradiation, and its effect was evaluated using a UV-vis spectrophotometer and ^1^H NMR. To acquire the UV-vis spectra, the AgTMPPS solution (concentration of 20 µmol/L) was mixed with NADH (125 µmol/L) in a quartz cell and irradiated by a laser (460 nm, 50 mw/cm^2^) for different durations. The absorbance changes at 339 nm were recorded to quantify the conversion level and kinetic data for conversion from NADH to NAD^+^. To acquire the ^1^H NMR spectra, NADH (4.5 mmol/L) and AgTMPPS (0.5 mmol/L) in a CD_3_OD/D_2_O (3/1) mixture were added to an NMR tube, and then continuously irradiated with a 460 nm laser for 10 min. A ^1^H NMR spectrometer (AVANCE Ⅲ 400 MHz, Bruker) was used to record the spectra at 310 K.

### 3.8. DFT Computation

All quantum chemical calculations were performed with the Gaussian 16 program package. The structures of the compounds under study were fully optimized using the B3LYP functional. The LANL2DZ basis set was used for the metal atoms, and the 6-311G(d) basis set was for the light atoms. The vibrational frequencies of the optimized structures were determined at the same level. The structures were characterized as local energy minima on the potential energy surface by verifying that all the vibrational frequencies were real. 

### 3.9. Antibacterial Performance Analysis

To characterize the antibacterial performance, the gram-positive MRSA strain (MRSA, ATCC 43300) was selected as the model. MRSA was purchased from ATCC, and an individual colony was collected from an LB agar plate stored at 4 °C and inoculated into a fresh LB broth. After overnight incubation at 37 °C and 220 rpm, the bacterial suspension was diluted at 1:200 and shaken for another 4 h to remain in the logarithmic growth phase before the experiments. The MRSA suspension was then stored at 4 °C. A laser system was used to induce a photodynamic effect and performed the in vitro antibacterial assay. The MIC and MBC values of TMPPS, AgTMPPS, and AgNO_3_ were determined. Sterilized compound solutions (initial concentration of 128 µmol/L) were sequentially diluted twice and mixed with the same amount of bacterial suspension (10^8^ CFU/mL). After co-culturing for 30 min at 37 °C, the mixtures were irradiated by a 460 nm laser (50 mw/cm^2^) for various durations and further incubated for 24 h. The MIC values were set as the lowest concentrations of the antimicrobial substance to prevent bacterial growth, from three independent experiments. The MBC values were measured by the dilution-in-broth method. The clear solutions were removed from wells and then added to fresh LB broth at 1:100. After incubation at 37 °C overnight, the MBC was considered the minimum drug concentration where no bacterial growth occurred, and three replicate experiments were performed.

The in vitro bacterial inhibition of the samples under different concentrations was determined by counting the colonies in the agar plate. AgTMPPS was dispersed in bacterial suspensions as described above, but MRSA had an OD_600_ value of 0.4. After 120 min of continuous laser irradiation, the bacterial suspensions were diluted 10^4^ times and inoculated onto fresh LB agar plates. The plates were cultured for 16 h at 37 °C, following which their images and colony counts were obtained. Three parallel species were prepared for the test. The results were analyzed using Prism 8 (Graph-Pad Software Inc., CA, USA), and the data are presented as the mean ± standard deviation (SD). The statistical significance was determined by ordinary one-way ANOVA tests with Tukey’s post-hoc test. The sample size (n) of the data is provided in the figure legend, and differences between groups are considered significant at *p* < 0.05. (**** *p* < 0.0001)

## 4. Conclusions

In conclusion we synthesized a novel Ag(II) water-soluble porphyrin, AgTMPPS, for synergistic bacterial sterilization against invading MRSA pathogens. The fabricated complex generated stable silver ions in high valence, released multiple abundant ^1^O_2_, and catalytically oxidized NADH under 460 nm irradiation. Systematic antibacterial experiments using AgTMPPS illustrated that the bactericidal ratio for highly concentrated MRSA (10^8^ CFU/mL) at a very low dosage (4 μmol/L) was 100%, making AgTMPPS a good antibacterial agent. Furthermore, DFT quantum chemical calculations indicated that the Ag(Ⅱ) ion fitted the porphyrin cavity well, and AgTMPPS was easier to transform to the excited state after light irradiation. This work highlights a dual-antibacterial strategy with a silver complex of a specific valency to provide novel routes for non-invasive therapy against resistant bacteria.

## Figures and Tables

**Figure 1 molecules-27-06009-f001:**
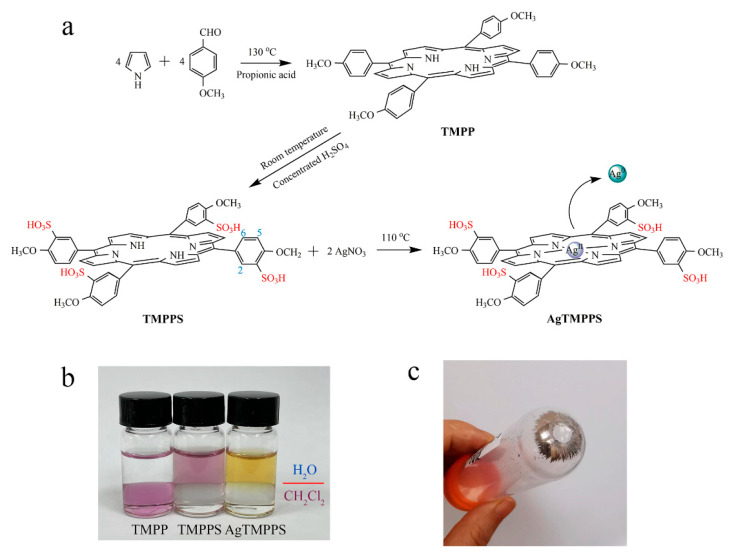
Synthesis of the silver-based porphyrin. (**a**) Schematic illustrating the fabrication of AgTMPPS. The numbers represent the positions of protons. (**b**) Solubility of porphyrins in two-phase media (H_2_O/CH_2_Cl_2_). (**c**) Precipitate after centrifugation of the reaction suspension.

**Figure 2 molecules-27-06009-f002:**
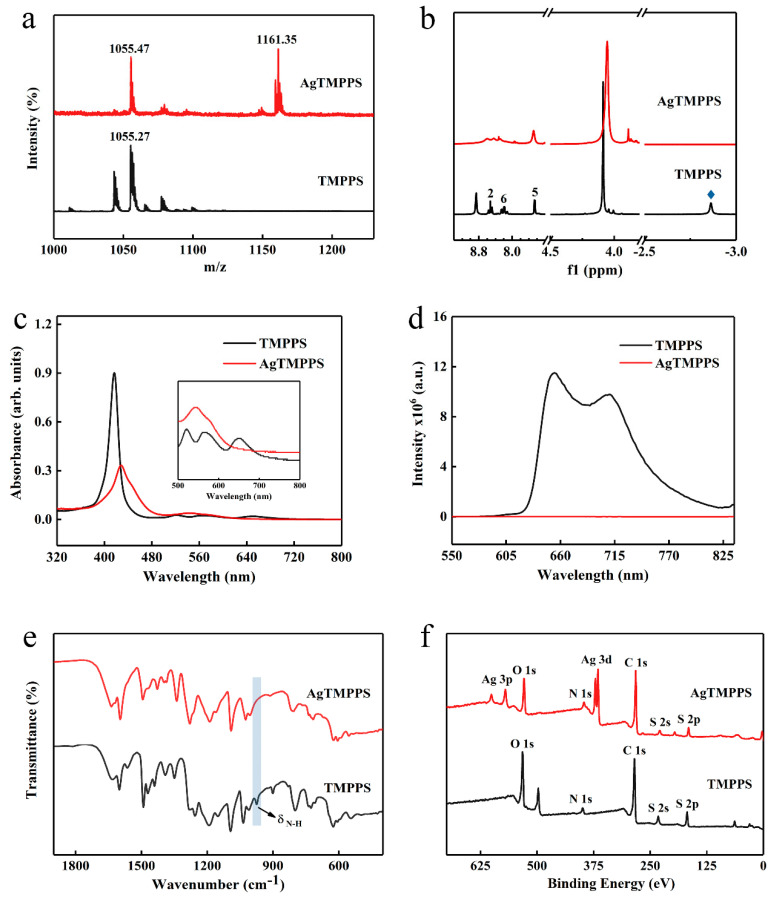
Characterization of TMPPS and AgTMPPS. (**a**) MALDI-TOF mass spectrum of TMPPS and AgTMPPS in ethanol. (**b**) 500 MHz ^1^H NMR spectrum of TMPPS and AgTMPPS. The number and blue rhombus represent the proton signals in different chemical environments (Figure 1a). (**c**) UV-vis absorption spectra of TMPPS and AgTMPPS in water. (**d**) Fluorescence emission spectra of TMPPS and AgTMPPS in water (λ_ex_ = 420 nm). (**e**) FT-IR spectra of TMPPS and AgTMPPS. (**f**) XPS survey scan spectra of TMPPS and AgTMPPS.

**Figure 3 molecules-27-06009-f003:**
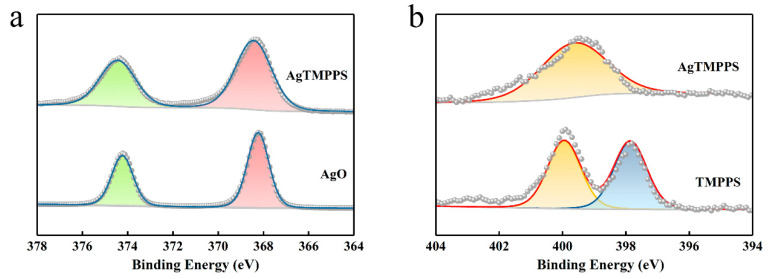
XPS narrow-scan spectra. (**a**) Ag 3d spectra of AgO and AgTMPPS. (**b**) N 1s spectra of TMPPS and AgTMPPS.

**Figure 4 molecules-27-06009-f004:**
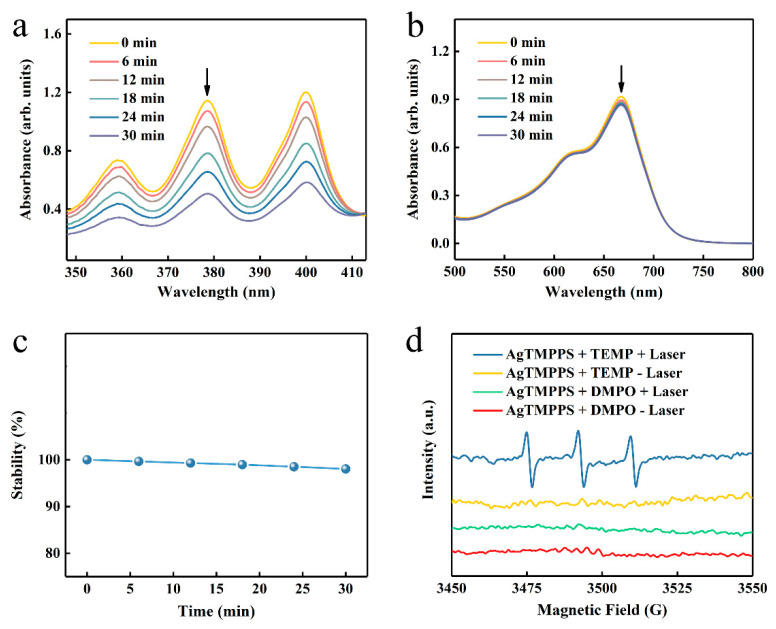
ROS generation by AgTMPPS under laser irradiation (460 nm, 0.05 W/cm^2^). (**a**) Time-dependent absorption of ABDA, suggesting the generation of ^1^O_2_ by AgTMPPS under laser irradiation. (**b**) Time-dependent degradation of MB by AgTMPPS under laser irradiation, suggesting no •OH detection. (**c**) Time-dependent stability of AgTMPPS under laser irradiation. (**d**) EPR spectra demonstrating ROS generation by AgTMPPS. (+) represents irradiation and (-) implies no irradiation. “Laser” implies irradiation at 460 nm (0.05 W/cm^2^).

**Figure 5 molecules-27-06009-f005:**
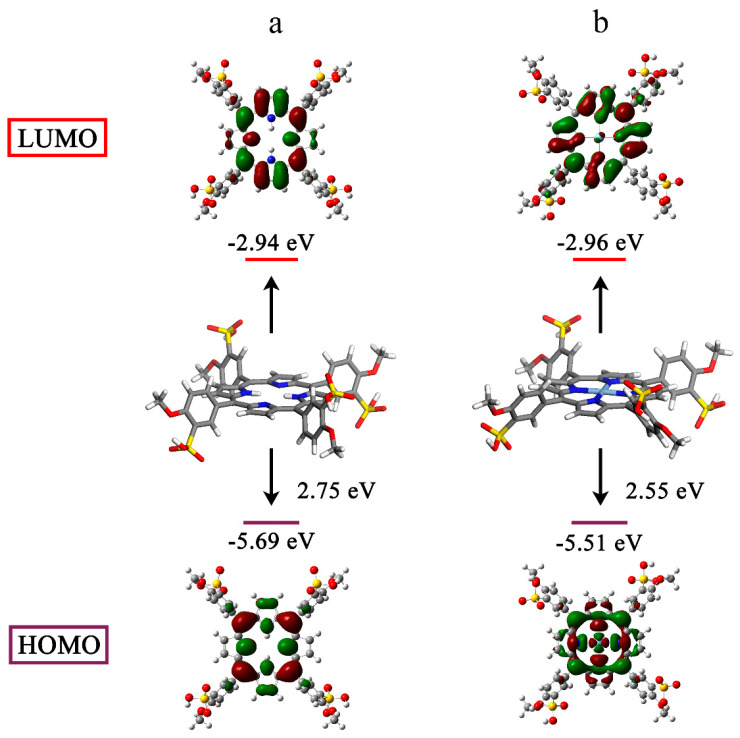
HOMO-LUMO profiles and energies of TMPPS (**a**) and AgTMPPS (**b**) determined by DFT. All calculations were performed at the DFT/TDDFT/B3LYP/6-311G(d)/LANL2DZ level of theory.

**Figure 6 molecules-27-06009-f006:**
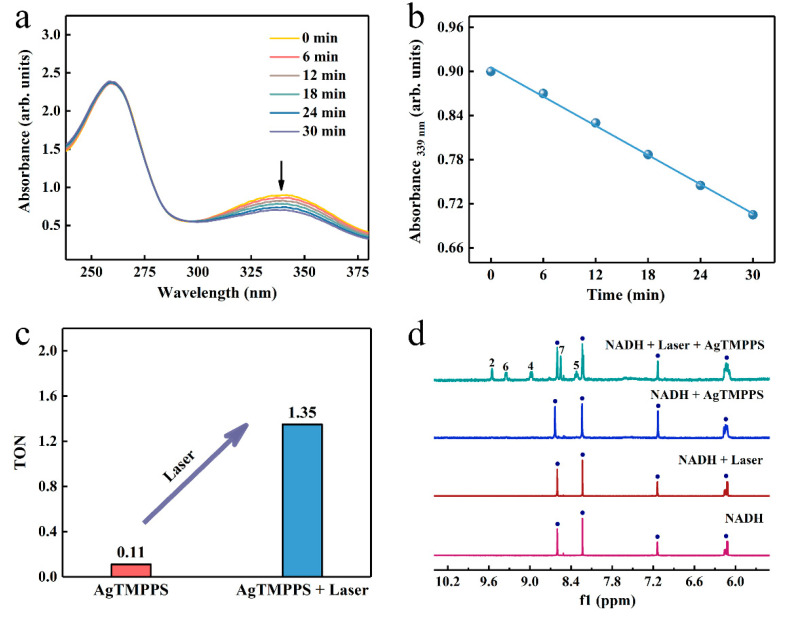
Photocatalytic oxidation of NADH by AgTMPPS and laser irradiation (460 nm, 0.05 W/cm^2^). (**a**) Reaction of AgTMPPS and NADH in ultrapure water under laser irradiation detected by UV-vis spectra. (**b**) Rate constant for NADH depletion as a function of the laser irradiation time in the presence of AgTMPPS. (**c**) TONs of NADH oxidation for the AgTMPPS and AgTMPPS + Laser groups. (**d**) Photocatalytic oxidation of NADH by AgTMPPS monitored by ^1^H NMR spectroscopy. The numbers denote the proton positions of NAD^+^ nicotinamide (Appendix A). The blue circles represent the signal peaks of NADH.

**Figure 7 molecules-27-06009-f007:**
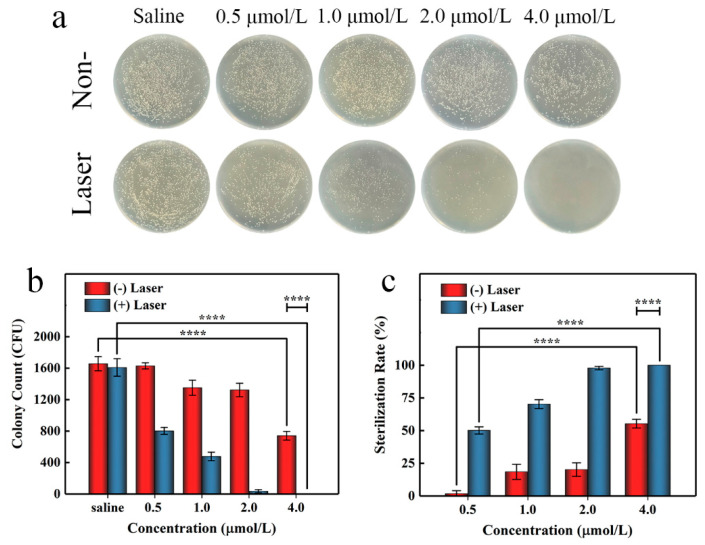
In vitro antibacterial performance of AgTMPPS against MRSA at different concentrations with or without 360 J/cm^2^ light irradiation (460 nm). (**a**) Photographs of the LB agar plate. (**b**) Corresponding CFU count. (**c**) MRSA sterilization rate. The asterisks indicate significant differences (*p* values: **** *p* < 0.0001). All values are expressed as mean ± SD, *n* = 3.

**Table 1 molecules-27-06009-t001:** Main electronic transitions, compositions of the low-lying electronically excited states, electronic excitation energies (eV), oscillator strengths (f), and the calculated possible fluorescence lifetimes (τ) of TMPPS and AgTMPPS ^a^.

Sample	Electronic Transitions	Compositions ^b^	Energy (eV)/Wavelength (nm)	f	τ (ns)
TMPPS	S_0_→S_1_	HOMO→LUMO (65.6%) HOMO − 1→LUMO + 1 (33.8%)	2.1631/573.18	0.0261	188.69
	S_0_→S_2_	HOMO→LUMO + 1 (62.0%) HOMO − 1→LUMO (37.6%)	2.2997/539.13	0.0339	128.53
	S_0_→S_4_	HOMO − 1→LUMO (59.2%) HOMO→LUMO + 1 (36.6%)	3.1082/398.89	1.7671	1.35
AgTMPPS	S_0_→S_1_	HOMO→LUMO + 1 (53.1%) HOMO→LUMO + 2 (27.8%)	1.5404/804.88	0.0007	13,872.97
	S_0_→S_2_	HOMO→LUMO (53.0%) HOMO→LUMO + 1 (27.9%)	1.5407/804.73	0.0007	13,867.57
	S_0_→S_4_	HOMO − 1→LUMO + 1 (37.3%) HOMO→LUMO + 1 (29.0%) HOMO→LUMO + 2 (14.8%)	1.8379/674.60	0.0088	775.19

^a^: Calculated by TDDFT//B3LYP/6-311G(d)/LANL2DZ PCM (water), based on the DFT//B3LYP/6-311G(d)/LANL2DZ optimized ground-state geometries. ^b^: Only the low-lying excited states and some allowed transitions are presented.

**Table 2 molecules-27-06009-t002:** MIC/MBC values of MRSA pathogens incubated with AgTMPPS and AgNO_3_ under different laser irradiation doses *.

Entry	Irradiation Dose (J/cm^2^)	MIC/MBC (μmol/L)	
		AgTMPPS	AgNO_3_
1	0	16/64	32/128
2	270	16/16	32/64
3	360	8/8	32/64
4	450	4/8	32/64

*: after 24 h incubation. All values are average of three replicates.

**Table 3 molecules-27-06009-t003:** Bacterial colony count of MRSA and the calculated sterilization rate for treatment with AgTMPPS at different concentrations ^a^.

Entry ^b^	AgTMPPS Concentration (μmol/L)	Colony Count (CFU)	Sterilization Rate (%)
1	Saline without light	1658 ± 90	-
2	0.5	1629 ± 39	1.71 ± 2.33
3	1.0	1351 ± 96	18.48 ± 5.77
4	2.0	1323 ± 86	20.18 ± 5.17
5	4.0	741 ± 55	55.29 ± 3.32
6	Saline with light	1609 ± 111	-
7	0.5	802 ± 44	50.14 ± 2.74
8	1.0	478 ± 54	70.29 ± 3.35
9	2.0	35 ± 21	97.82 ± 1.29
10	4.0	0	100

^a^: The experiments were conducted under 360 J/cm^2^ irradiation or without irradiation, and again after 16 h of incubation. All values are expressed as mean ± SD, *n* = 3. ^b^: Entries 1–5 show results for treatment without irradiation, and entries 6–10, with irradiation.

## Data Availability

The data presented in this study are available on request from the corresponding authors.

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
