# Peer review of "Synthesis of a Rare Water-Soluble Silver(II)/Porphyrin and Its Multifunctional Therapeutic Effect on Methicillin-Resistant Staphylococcus aureus"

_molecules, 2022, doi:10.3390/molecules27186009_

Round 1
Reviewer 1 Report
In this work, the authors reported a water-soluble silver(II)/porphyrin was synthesized and characterized by various methods. They also performed the antibacterial experiments to evaluate the sterilization performance of AgTMPPS. Antibacterial experiments on methicillin-resistant Staphylococcus aureus (MRSA) revealed that the combined action of Agâ…¡ ions and PDT could endow AgTMPPS with an excellent bactericidal ratio. Authors also provided theoretical calculations for understanding the antibacterial properties of TMPPS and AgTMPPS. The following issues still to be addressed. I hence recommend a major revision.
1. The authors synthesized AgIITMPPS via a classic coordination reaction between TMPPS ligand and AgINO3. What is the mechanism of the formation of Ag2+ in the core of AgTMPPS? What about the stability AgIITMPPS molecules?
2. The authors confirmed that the precipitate is to be elemental silver by the XPS tests, but the peaks position of Ag-3d spectra in Figure 3a and Figure S1 are the same. How can the author distinguish the Ag2+ and elemental Ag through the XPS?
3. The porphyrin TMPPS as a PDT agent, the singlet oxygen can be produced by the TMPPS under 460 nm irradiation?
4. The authors mentioned the MIC and MBC for solutions of the TMPPS, they are more than 128 μmol/L. How this value obtained?
5. The authors demonstrated that AgTMPPS can photocatalytically oxidize the 1,4-dihydronicotinamide adenine dinucleotide to NAD+. This experiment and antibacterial experiments on methicillin-resistant staphylococcus aureus seem to be unrelated. What is the purpose of the photocatalytically oxidized test?
Reviewer 2 Report
The author skillfully synthesized a porphyrin photosensitizer with an unusual Ag(II) oxida-tion state, and characterized the compound by various means to prove the existence of Ag2+. AgTMPPS has obvious inhibitory effect on methicillin-resistant Staphylococcus aureus in antibacterial experiments. The PDT reaction was proved by photocatalytic oxidation of 1, 4-dihydronicotinamide adenine dinucleotide to NAD + by AgTMPPS. Therefore, this manuscript meets the requirements of journals. In my opinion. This article is acceptable after major changes.
1. The mass spectral data of compounds TMPPS and AgTMPPS are labeled incorrectly in Figure 2a, and the illustration of Figure 4B should be modified
2. The author uses dichloromethane and water to prove that the synthesized AgTMPPS is water-soluble (Fig. 1b). but I have a question. What is the solubility of AgTMPPS in water?
3. The singlet oxygen produced by AgTMPPS under illumination was detected by ABDA probe, but the singlet oxygen yield was not calculated. I request that the authors make additional clarifications on this.
4. The fluorescence emission of TMPPS and AgTMPPS in water was measured at wavelengths between 600 and 800nm (Fig. 2d). At the same time, density functional theory is used to explain the reason, which is due to the transition of free porphyrin in S0 ~ S4 (Table 1). However, the calculated fluorescence emission wavelength should be about 400nm. Therefore, I do not agree with the author's explanation and need further explanation from the author.
5. In Vitro Antibacterial Performance of the Silver-Based Porphyrin against MRSA, the MIC/MBC values of TMPPS are all greater than 128 μmol/L. Is this data obtained by testing? The author is required to provide MIC and MBC values of TMPPS under different laser irradiation doses.
Reviewer 3 Report
-just for readers, the authors should explain the differences between antibacterial and bactericidal means?
-The authors can cite the "Light-driven antimicrobial therapy of palladium porphyrins and their chitosan immobilization derivatives and their photophysical-chemical properties" Dyes and Pigments 203 (2022) 110313 since it fit this study's concept
Round 2
Reviewer 2 Report
The author answered my questions seriously and gave explanations, which I was very satisfied with. Therefore, there is no further comments.